# Implementation and evaluation of an antimicrobial stewardship programme in companion animal clinics: A stepped-wedge design intervention study

Nonke E. M. Hopman[1], Lützen Portengen[2], Marlies E. J. L. Hulscher[3], Dick J. J. Heederik[2], T. J. M. Verheij[4], Jaap A. Wagenaar[1,5], Jan M. Prins[6], Tjerk Bosje[7], Louska Schipper[2], Ingeborg M. van Geijlswijk[2,8], Els M. Broens[1]*

1 Department of Infectious Diseases and Immunology, Faculty of Veterinary Medicine, Utrecht University, Utrecht, the Netherlands, 2 Institute for Risk Assessment Sciences, Utrecht University, Utrecht, the Netherlands, 3 Scientific Center for Quality of Healthcare (IQ healthcare), Radboud Institute for Health Sciences, Radboud University Medical Center, Nijmegen, the Netherlands, 4 Julius Center for Health Sciences and Primary care, University Medical Center, Utrecht, the Netherlands, 5 Wageningen Bioveterinary Research, Lelystad, the Netherlands, 6 Department of Internal Medicine, Amsterdam University Medical Center, University of Amsterdam, Amsterdam, the Netherlands, 7 Medical Center for Animals, Amsterdam, the Netherlands, 8 Pharmacy Department, Faculty of Veterinary Medicine, Utrecht University, Utrecht, the Netherlands

* e.m.broens@uu.nl

## Abstract

### Background

To curb increasing resistance rates, responsible antimicrobial use (AMU) is needed, both in human and veterinary medicine. In human healthcare, antimicrobial stewardship programmes (ASPs) have been implemented worldwide to improve appropriate AMU. No ASPs have been developed for and implemented in companion animal clinics yet.

### Objectives

The objective of the present study was to implement and evaluate the effectiveness of an ASP in 44 Dutch companion animal clinics. The objectives of the ASP were to increase awareness on AMU, to decrease total AMU whenever possible and to shift AMU towards 1st choice antimicrobials, according to Dutch guidelines on veterinary AMU.

### Methods

The study was designed as a prospective, stepped-wedge, intervention study, which was performed from March 2016 until March 2018. The multifaceted intervention was developed using previous qualitative and quantitative research on current prescribing behaviour in Dutch companion animal clinics. The number of Defined Daily Doses for Animal (DDDAs) per clinic (total, 1st, 2nd and 3rd choice AMU) was used to quantify systemic AMU. Monthly AMU data were described using a mixed effect time series model with auto-regression. The effect of the ASP was modelled using a step function and a change in the (linear) time trend.

**Data Availability Statement:** The data underlying this study are owned by private companion animal clinics, who have restricted their public sharing

because the data contain confidential information (i.e. patient and owner information). Data was shared with the researchers using a data user agreement stating that the data would be used anonymous by the authors for scientific purposes only. The anonymized dataset used for the modelling is available from the Utrecht University YODA Data management system (https://dgk.yoda.uu.nl/; contact via corresponding author or yoda@uu.nl) for researchers who meet the criteria for access to confidential data.

**Funding:** This work was supported by ZonMw (Netherlands Organisation for Health Research and Development, The Hague, the Netherlands), project number 205300003. The funder had no role in study design, data collection and analysis, decision to publish, or preparation of the manuscript.

**Competing interests:** The authors have declared that no competing interests exist.

## Results

A statistically significant decrease of 15% (7%-22%) in total AMU, 15% (5%-24%) in 1st choice AMU and 26% (17%-34%) in 2nd choice AMU was attributed to participation in the ASP, on top of the already ongoing time trends. Use of 3rd choice AMs did not significantly decrease by participation in the ASP. The change in total AMU became more prominent over time, with a 16% (4%-26%) decrease in (linear) time trend per year.

## Conclusions

This study shows that, although AMU in Dutch companion animal clinics was already decreasing and changing, AMU could be further optimised by participation in an antimicrobial stewardship programme.

## Introduction

The increase of antimicrobial resistance (AMR) is recognised as a threat for modern medicine and public health [1]. To help control AMR, responsible use of antimicrobials (AMs) is warranted and a decrease in inappropriate use of AMs is necessary, both in human and veterinary medicine [1–3].

In human medicine, the term antimicrobial stewardship programme (ASP) generally refers to specific programmes or series of interventions to monitor and direct antimicrobial use (AMU) at the hospital or primary care level [4–7]. In veterinary medicine, it usually encompasses numerous elements of improved AMU (e.g., increasing awareness of (inter)national practice guidelines, use of diagnostic microbiology and use of alternatives to AMs) and it is often associated with country-wide surveillance of AMU and development of (inter)national guidelines on AMU [8].

In Dutch food producing animals, a combination of compulsory and voluntary actions resulted in a 64% reduction in AMU (between 2008 and 2017). A decrease in resistance rates was observed as well [9–12]. Just since the end of 2011 onwards, more attention is being paid to AMU in companion animals. Legislation (2013) on mandatory susceptibility testing for veterinary use of 3rd choice AMs also holds for companion animals [13]. The Royal Dutch Veterinary Association promotes the use of guidelines on AMU as well.

A survey on prescription data of 68 companion animal clinics in the Netherlands (during 2009–2011) showed that the use of 3rd generation cephalosporins and fluoroquinolones (i.e. highest priority critically important antimicrobials for human medicine according to the World Health Organisation (WHO)) accounted for 18% of total AMU, based upon the number of Defined Daily Doses for Animals [14,15]. During the past years, AMU in Dutch companion animal clinics has been decreasing (with 19% when comparing 2012 to 2014) [16]. However, especially with regard to the (sub)classes of AMs used, there is still room left for improvement. According to Dutch classification of veterinary AMU, 2nd choice AMs (i.e. mainly aminopenicillins and 1st and 2nd generation cephalosporins) still accounted for 43% of total AMU and 3rd choice AMs (i.e. 3rd generation cephalosporins and fluoroquinolones) for 8% of total AMU in 2014 [16].

Antimicrobial prescribing behaviour can only be improved if interventions are attuned to the specific situation and the target group, and factors influencing antimicrobial prescribing are taken into account [17,18]. Qualitative research in Dutch companion animal clinics

indicated that antimicrobial prescribing behaviour is influenced by four main categories of factors: veterinarian-related factors, patient-related factors, treatment-related factors and contextual factors [19]. These categories of factors were taken into account when the intervention elements of the present study were developed. The aim of this study was to implement and evaluate the effectiveness of such a customised ASP, aiming to improve antimicrobial prescribing, in 44 Dutch companion animal clinics.

## Materials and methods

### Study design

The Antimicrobial Stewardship and Pets-study (ASAP) was designed as a prospective, stepped-wedge intervention study aiming to optimise antimicrobial prescribing in Dutch companion animal clinics by implementing an antimicrobial stewardship programme. The intervention study was performed from March 2016 until March 2018.

### Time schedule

Clinics, divided into four clusters, were offered all separate intervention elements of the ASP. The period considered as the actual "intervention period" comprised 12 months: from start of implementation of the ASP (i.e. filling in patient evaluation forms) up to 4–5 months after the feedback meeting (i.e. when clinics started filling in patient evaluation forms for the second time). Time schedule of the applied stepped-wedge design is shown in Fig 1. The first cluster of clinics was contacted in December 2015, their intervention period started in March 2016. The intervention period of the last cluster started in January 2017. The clinics were grouped into four clusters based on their geographic location.

### Participating clinics

**Sample size.** The study aimed at including at least 40 clinics. This number was based on a power calculation which indicated that with 40 clinics a minimal change of 8% in mean total AMU could be detected with a power of 0.80 (= beta) and a significance of 0.05 (= alpha) over a one-year period. Calculations were based on AMU data at clinic level from a previous study conducted in 68 Dutch companion animal clinics [15].

**Clinic selection.** Companion animal clinics were approached for participation using the database of the Royal Dutch Veterinary Association (KNMvD) containing all Dutch veterinary

| | 2015 | 2016 | | | | 2017 | | | | 2018 |
|---|---|---|---|---|---|---|---|---|---|---|
| | Dec | Jan-March | Apr-June | Jul-Sep | Oct-Dec | Jan-March | Apr-June | Jul-Sep | Oct-Dec | Jan-March |
| Cluster 1 | ****** | 11* | | | F | | ****** | | | |
| Cluster 2 | | ****** | 11* | | | F | | ****** | | |
| Cluster 3 | | | ****** | 10* | | | F | | ****** | |
| Cluster 4 | | | | ****** | 12* | | | F | | ****** |

****** = retrieving AM prescription data retrospectively from the PMS (before and after participation in the ASP)

\* Start of implementation of the ASP, and the number of participating clinics per cluster

F: Feedback meeting

**Fig 1. Time schedule of the applied stepped-wedge design.** The time schedule indicates the start and the duration of the period considered as the "intervention period" for the four separate clusters of participating clinics.

clinics. The clinics were sampled based upon previous shown interest to participate in a study on optimisation of AMU in companion animal clinics and on geographic location.

Clinics were invited by e-mail, followed a week later by a phone call to answer questions and to arrange a visit. Ultimately, clinics were only included when the Practice Management System (PMS) appeared to be able to provide information on antimicrobial prescription data specified for companion animals on a monthly basis. Clinics treating companion animals only and mixed clinics (i.e. clinics treating companion animals and non-companion animals, but with separated companion and non-companion animal prescription data) were included.

Clinics were offered a financial compensation, which was based upon estimated time investment per clinic and a standard hourly tariff for veterinarians. Educational training included in the ASP was accredited as professional continuing education for participating veterinarians.

The study was exempt from ethical approval as no animal experiments were involved. Participating veterinarians remained fully autonomous in their daily practice. Before enrolment, all clinics received written and oral information on the purpose of the study. Every clinic signed an informed consent to confirm their commitment to participate and to give permission for the use of their patient data for research purposes after anonymisation.

## Applied intervention approach

A stewardship programme to optimise AMU was developed based upon previous qualitative research [19] and field experiences from co-authors involved in human medicine. The objectives of the ASP were to increase awareness on AMU, to decrease total AMU whenever possible and to shift AMU towards 1st choice agents, which is according to current guidelines on AMU. Cues from the RESET Model to change human behaviour were used; Rules & regulations, Education & information, Social pressure, Economics, and Tools [20,21]. A Support-Team (S-Team) was assembled, in the analogy of the human Antibiotic Stewardship-Teams (A-team) [22,23]. The S-team included a veterinary microbiologist, a veterinary specialist in internal medicine of companion animals, a veterinary pharmacologist, a hospital pharmacist and the project leader. The S-Team members were involved in the different elements of the ASP (Table 1).

The Dutch classification [26] of veterinary AMU (Table 2), current Dutch guidelines (on otitis externa, urinary tract infections and skin infections) and formulary on veterinary AMU, and legislation on mandatory susceptibility testing for veterinary use of 3rd choice AMs were used as treatment standards during the ASP [13,24–26].

## Data collection and management

Participating clinics supplied clinic population data and monthly antimicrobial veterinary medicinal product (AVMP) prescription data. Information on the composition of the clinic's animal population (represented by the number of dogs, cats and rabbits attending the clinic at least once in a specified 3-year period) and monthly AM prescription data were retrieved retrospectively from the PMS, once before participation in the ASP and once after participation in the ASP.

## Outcome measures

The primary outcome measure was total AMU. AMU was further classified into 1st, 2nd and 3rd choice AMU (Table 2).

Systemic AMU was quantified as described and discussed in detail in previous study [16] and is comparable with the Defined Daily Dose Animal ($DDDA_{VET}$), a measure suggested by

**Table 1. Separate intervention elements as offered during the ASP, including when they were offered, who were involved and the estimated time investment for participants.**

| Intervention element | When | Who were involved | Estimated time investment |
|---|---|---|---|
| 1) Filling in (preferably) 100 patient evaluation forms per clinic; to reflect on own AM prescribing behaviour | At the start of the intervention period & at the end | Veterinarians | 2–5 minutes per evaluation form |
| 2) Post educational training 1; on AMR, international and national regulations, and guidelines on responsible AMU | Month 1 | Veterinarians and 2 S-Team members | 2.5 hours |
| 3) Exercise to write down own AM prescribing behaviour; to compare it with current guidelines and to discuss it with colleagues | Between post educational training 1 & 2 | Veterinarians within the same clinic | 2 hours |
| 4) Post educational training 2; on behavioural change and communication skills towards companion animal owners | Month 4 | Veterinarians, veterinary nurses, 2 S-Team members and 1 communication trainer | 2.5 hours |
| 5) Commitment form; to sign within the clinic, committing to use AMs responsibly | After post educational training 2 | Veterinarians and veterinary nurses | 0.5 hour |
| 6) Benchmarking of quantitative AMU data | During post educational training 1 and the feedback meeting | Veterinarians and S-Team members | |
| 7) Information leaflet for companion animal owners on responsible AMU and AMR | During participation in the intervention programme | Veterinarians and veterinary nurses | |
| 8) Asking questions to the S-team members, on AMU and AMR, via email or phone call | During participation in the intervention programme | Veterinarians and S-Team members | |
| 9) Feedback meeting; every clinic was visited once, clinic-based feedback was given on all gathered data on AMU (1, 3 and 6). Clinic specific AMU objectives were defined, questions were answered and topics on AMU and AMR were discussed | Month 8 | Veterinarians and 2 S-Team members | 2–3 hours |

the European Surveillance of Veterinary Antimicrobial Consumption group (ESVAC) [27,28]. In summary, a $DDDA_{CLINIC}$ of 0.25 per month means that the average dog, cat and rabbit in the clinic was treated with antimicrobials for a 0.25 day per month. Per clinic, total $DDDA_{CLINIC}$ was calculated per month and specified with $1^{st}$, $2^{nd}$ and $3^{rd}$ choice AMU. In the present study, only systemic (i.e. oral or parenteral) AMU was described. AMs applied topically were excluded from analyses.

Mean absolute AMU numbers are presented for 12–24 months and 0–12 months prior to the start of the intervention period and for a period of 12 months considered as the actual intervention period, taking one month transition time into account.

## Statistical analysis

A mixed effect time series model was used to describe monthly AMU from 12 months before until 12 months after the introduction of the ASP, that allowed for a linear trend over time, while seasonal patterns were modelled using Fourier (sine- and cosine-) terms. AMU appeared

**Table 2. Classification of veterinary AMU according to Dutch policy on veterinary AMU [26].**

| Classification | Reasoning | Main classes of AMs |
|---|---|---|
| $1^{st}$ choice | Empirical therapy; Do not select for (to current knowledge), nor are specifically meant for treatment of ESBL-producing micro-organisms. | Tetracyclines, nitroimidazoles, narrow-spectrum penicillins, trimethoprim, sulfonamides, lincosamides and phenicols. |
| $2^{nd}$ choice | All AMs not classified as $1^{st}$ or $3^{rd}$ choice AMs; Use of these AMs might select for ESBL-producing bacteria or is specifically indicated in case of an ESBL-infection. | Aminopenicillins (with/without beta-lactamase inhibitors), $1^{st}$ and $2^{nd}$ generation cephalosporins, aminoglycosides and colistin. |
| $3^{rd}$ choice | A selection of Highest Priority Critically Important AMs for human medicine according to WHO; By Dutch law restricted to use only in individual animals and after culture and susceptibility testing. | Fluoroquinolones, $3^{rd}$ and $4^{th}$ generation cephalosporins. |

to follow an approximate log-normal distribution and therefore log-transformed AMU-data were used as the dependent variable. As a result, presented models estimate geometric mean (GM) AMU. Geometric mean ratios (GMRs; the ratio of two GMs) are used to quantify effects (e.g., the ratio of the GM during the intervention period to that before the intervention period).

The effect of the intervention was modelled using a step function and by modelling a change in the (linear) time trend. For each clinic, a dummy variable was included to indicate the month the ASP was introduced, because AMU in that month could not be unambiguously assigned to either the pre- or post-intervention period (transition period).

Heterogeneity in baseline AMU, time trends, seasonal patterns, and intervention effects across different clinics were modeled using (correlated) random effects. Short-term time series dynamics were accounted for by an auto-regressive (AR1) structure and the residual variance was allowed to be different for each clinic.

The estimated average intervention effects (i.e. the stepwise change and change in time trend) across clinics are reported. The model was used to evaluate the overall effect of the ASP for total, 1st, 2nd, and 3rd choice AMU separately. Effects are expressed as GMRs and (alternatively) as proportional decreases in use. $P_{WALD}$ values are used to indicate whether the separate coefficients are significantly different from 0, $P_F$ values are used to indicate the significance of the overall intervention effect.

SAS (SAS 9.4, SAS Institute, Inc. Cary, NC, USA) was used to organise the data and the nlme package (version 3.1) in R (version 3.5) was used to perform the statistical analyses.

## Results

### Participating clinics

In total, 54 clinics were contacted to participate. Six of these clinics were not willing to participate and four clinics were excluded, because their PMS appeared not suitable to provide monthly prescription data. Finally, 44 clinics were included in the study.

Table 3 provides a summary of characteristics of participating clinics. All clinics provided AMU data prior to the introduction of the stewardship programme for a minimum of 25 months, except for one clinic, that provided data for only 13 months prior to the introduction of the ASP.

Data of 41 clinics were included in the data analysis. Three clinics were excluded from data analysis. One of these clinics was lost to follow-up (i.e. no response was received when (repeatedly) trying to retrieve the AM prescription data from the PMS after participation in the ASP) and two clinics had substantial changes in their clinic's animal composition (i.e. one clinic

**Table 3. Mean (range) or distribution of characteristics of 44 clinics participating in the ASP.**

| Characteristic (number of clinics = 44) | Mean (range) |
|---|---|
| Number of dogs | 2151 (14–5353) |
| Number of cats | 1910 (350–5113) |
| Number of rabbits | 271 (0–797) |
| Number of veterinarians treating companion animals | 2.7 (1–8) |
| Mean work experience per clinic (years) | 16.2 (5.8–34) |
| **Characteristic (number of clinics = 44)** | **Distribution** |
| Companion animals only versus mixed-animal clinics | 40 / 4 |
| Urban, rural or urban-rural | 29 / 14 / 1 |

**Table 4. Mean total, 1st, 2nd and 3rd choice AMU (in numbers of DDDA/month and percentage of total AMU) in participating clinics, before and during participation in the antimicrobial stewardship programme (ASP).**

| Classification of antimicrobials[1] | Pre-ASP period (12–24 months) | Pre-ASP period (0–12 months) | During participation in the ASP (2–13 months) |
|---|---|---|---|
| First choice (% of total) | 0.059 (44.1%) | 0.060 (45.5%) | 0.066 (57.8%) |
| Second choice (% of total) | 0.064 (47.6%) | 0.063 (48.0%) | 0.045 (39.2%) |
| Third choice (% of total) | 0.011 (8.3%) | 0.009 (6.5%) | 0.003 (3.0%) |
| Total DDDA per month (SD) | 0.134 | 0.132 | 0.114 |

[1] = according to Dutch policy on veterinary AMU (Table 2)

closed and one clinic opened an extra location with the same PMS making AMU calculations unreliable).

## Outcomes

**Mean, absolute AMU (total, 1st, 2nd and 3rd choice AMU).** Mean total AMU was 0.134 and 0.132 DDDA/month, respectively, in the two years prior to implementation of the ASP and decreased to 0.114 DDDA/month during the period considered as the intervention period (Table 4). Similar decreasing trends were seen for 3rd choice AMU (i.e. fluoroquinolones and 3rd generation cephalosporins) and 2nd choice AMU (i.e. mainly aminopenicillins, with and without clavulanic acid). AMU shifted towards more 1st choice AMU.

**Intervention effect.** As a result of participation in the ASP, a stepwise decrease was estimated for total, 1st and 2nd choice AMU of 15% (95% CI: 7%-22%; p<0.01), 15% (95% CI: 5%-24%; p<0.01) and 26% (95% CI: 17%-34%; p<0.01) respectively. No statistically significant effect was estimated for 3rd choice AMU. The change in (linear) time trend was statistically significant for total AMU only with an additional 16% decrease over the year (95% CI: 4%-26%; p = 0.01) (Table 5).

As absolute figures for 1st choice AMU were increasing (not visible from Fig 2A–2D, because the trend before the ASP was set at 100%), the net effect of the stepwise decrease in 1st choice AMU is a smaller increase in use than was expected based upon the pre-intervention time trend for 1st choice AMU.

Although AMU decreased in most clinics, there were considerable differences in estimated intervention effects between clinics (Fig 2A–2D).

**Table 5. Stepwise change and change in (linear) trend of total, 1st, 2nd, and 3rd choice AMU.**

| Classification of antimicrobials[1] | | GMR (95% CI) | % Decrease | P_WALD | P_F |
|---|---|---|---|---|---|
| **First choice** | Stepwise change in use | 0.85 (0.76–0.95) | 15% (5% to 24%) | <0.01 | <0.01 |
| | Change in (linear) trend (/year) | 0.92 (0.74–1.13) | 8% (-13% to 26%) | 0.41 | |
| **Second choice** | Stepwise change in use | 0.74 (0.66–0.83) | 26% (17% to 34%) | <0.01 | 0.01 |
| | Change in (linear) trend (/year) | 0.85 (0.69–1.04) | 15% (-4% to 31%) | 0.12 | |
| **Third choice** | Stepwise change in use | 0.94 (0.72–1.23) | 6% (-23% to 28%) | 0.66 | 0.62 |
| | Change in (linear) trend (/year) | 0.78 (0.46–1.32) | 22% (-32% to 54%) | 0.35 | |
| **Total** | Stepwise change in use | 0.85 (0.78–0.93) | 15% (7% to 22%) | <0.01 | <0.01 |
| | Change in (linear) trend (/year) | 0.84 (0.74–0.96) | 16% (4% to 26%) | 0.01 | |

Reported effects are averaged estimates of 41 participating clinics, from a random effects model that includes a (linear) time trend and seasonal effects, and allows for heterogeneity of effects between clinics and residual auto-correlation. Effects are expressed as GMRs and (alternatively) as proportional decreases in use.

[1] = according to Dutch policy on veterinary AMU (Table 2)

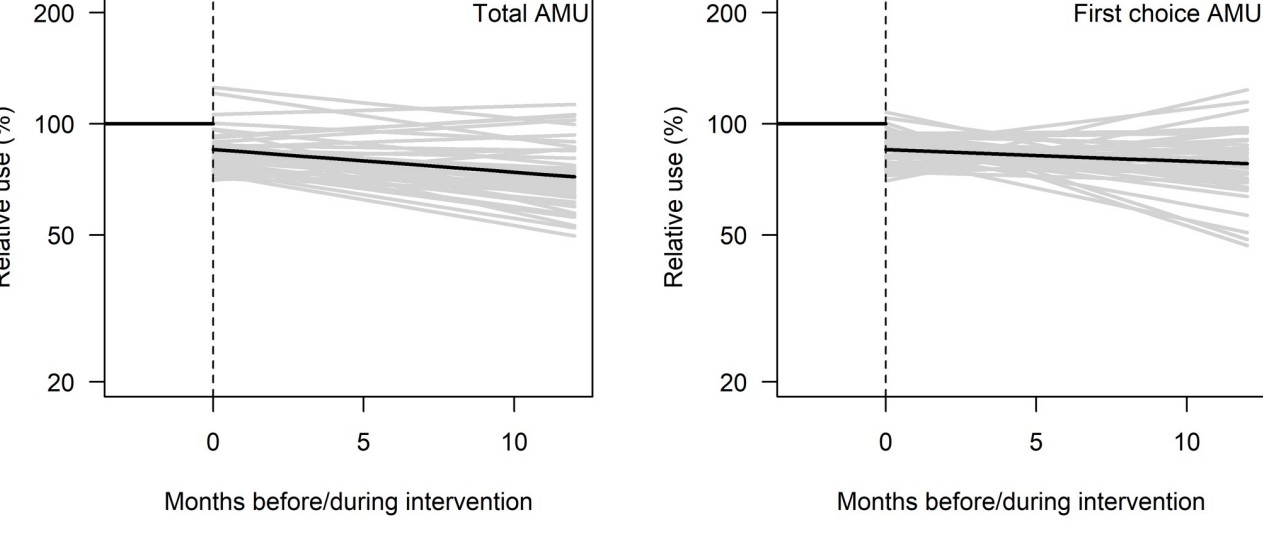

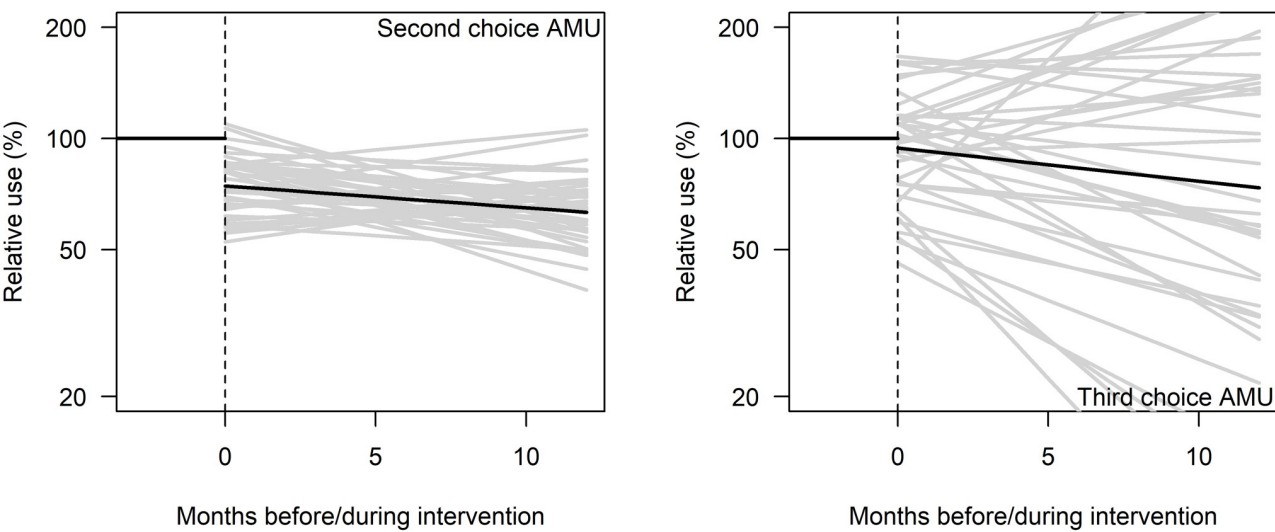

**Fig 2. Average and clinic-specific effects of the ASP on total, 1ˢᵗ, 2ⁿᵈ and 3ʳᵈ choice AMU.** Combined effect of participation in the ASP (stepwise change & change in time trend) are shown for the average effect (black) and for each individual clinic (grey) after standardisation to the estimated AMU before the intervention period (as 100%).

## Discussion

A strong and statistically significant decrease in total, 1ˢᵗ and 2ⁿᵈ choice AMU was observed in a sample of Dutch companion animal clinics, with a shift towards use of 1ˢᵗ choice antimicrobials, that was attributed to participation in an antimicrobial stewardship programme (ASP). The change in total AMU became more prominent over time after introduction of the ASP.

The results of the statistical model indicate that 1ˢᵗ choice AMU decreased. However, absolute 1ˢᵗ choice AMU increased during the intervention period. Therefore, after introduction of the ASP, the net effect was a less pronounced increase in use of 1ˢᵗ choice AMs than was expected. The statistical model used to estimate the intervention effect, assumed a linear,

random time trend per clinic and a stepwise reduction in AMU after adjustment for seasonal effects. The stepwise effect was observed in most clinics. No statistically significant effect on $3^{rd}$ choice AMU could be attributed to participation in the ASP. This is likely explained by the fact that $3^{rd}$ choice AMU was already reduced to a low level (0.009 DDDA/month, 0–12 months before implementation of the ASP) in the years preceding the ASP. Therefore, a further decrease as result of the ASP is difficult to demonstrate.

The goal of the ASP was to increase awareness on AMU, to decrease total AMU whenever possible and to shift AMU towards $1^{st}$ choice AMs, according to Dutch guidelines on veterinary AMU [24,26]. The observed changes in the clinics participating in the ASP are in line with the goal of the ASP and the Dutch guidelines on veterinary AMU, and are therefore considered relevant.

A strength of the design of the present study is that repeated monthly measurements per clinic were involved, which allowed to control intervention effects for baseline levels and ongoing time trends. By starting the ASP at different timepoints for the four different clusters of clinics, the probability that the overall effect was influenced by external events was minimised (e.g., increased attention on responsible AMU in general) [6].

In general, measures to evaluate the effect of an ASP can be divided into four main categories: patient outcomes, unintended consequences of AMU (e.g., adverse effects and emergence of AMR), AMU and costs, and process measures [29]. As a possible limitation of present study, it could be argued that overall AMU is a non-specific measure without information on appropriateness of AM therapy or patient outcomes, as is the case for looking at $1^{st}$, $2^{nd}$ and $3^{rd}$ choice AMU [6,29–31]. Moreover and more important, an increase in quality of AMU can be reached without a reduction in AMU (or even with an increase in AMU), e.g., by using more $1^{st}$ choice AMs instead of $2^{nd}$ or $3^{rd}$ choice AMs, or by using better dosing [6,32,33]. On the other hand, as any use of AMs selects for AMR, any reduction of AMU that can be achieved by improving adherence to current guidelines (by definition appropriate AMU) is an advantage, as is the case for using $1^{st}$ instead of $2^{nd}$ or $3^{rd}$ choice AMs. The persistence of the effect during the follow-up period of 12 months (especially of total AMU with a significant change in linear time trend) suggests sustainability of the changes in AMU. However, repeated AMU-measurements in the nearby future are needed to evaluate the sustainability over a longer period of time.

A second limitation is the fact that participating clinics were contacted approximately 2–3 months before actual start of the ASP. This could have led to a change in AM prescribing behaviour already, because clinics knew their AM prescribing behaviour would be monitored. These 2–3 months are part of the baseline measurement period. As a result, the intervention/ ASP effect could have been slightly diminished [6,34]. Another potential weakness of the stepped-wedge design is contamination of the interventions [6]. Information, insights or effects from clinics already having started the ASP could have influenced clinics still in the baseline period. Because participating clinics were clustered based on their geographic location, this effect was expected to be minimal, but could not be excluded.

The representativeness of the participating clinics for the whole country might be questioned. These 44 clinics were not randomly selected, but selection was based upon willingness to participate. It is possible that participating clinics already had a more responsible attitude towards AMU and had more motivation to change their AM prescribing behaviour compared to other not-participating clinics. On the other hand, results of the present study might also be regarded as a proof of principle. If even in clinics that already had an interest in responsible AMU, optimisation of AMU could be attained, clinics with less interest in responsible AMU might be able to change even more. However, and irrefutably, it will be harder to change

behaviour of veterinarians who do not believe that responsible AMU is a desirable or necessary behaviour [19,20,35].

Antimicrobial prescribing behaviour is influenced by many factors. Multifaced interventions, attuned to the specific setting and influencing factors of AMU are advised to optimise AM prescribing behaviour [7,8,18,19,36–38]. The present ASP was based upon a previous qualitative study and the RESET model, containing the most important cues to change human behaviour [19,20]. Only 'Economics', covering profits and costs, bonuses and penalties, as a factor influencing prescribing behaviour was not addressed directly in the ASP. However, the importance of economics was discussed (e.g., the difference between earning money because of prescribing AMs versus performing further diagnostics). Besides, clinics were aware of possible inspections by the Dutch Food and Consumer Product Safety Authority (NVWA) of the Dutch Government, possibly leading to financial penalties in case of prescribing 3rd choice AMs without culture and susceptibility testing.

The present study showed that the developed ASP was effective in reducing and refining AMU in the participating clinics. An evaluation survey among participants and a stakeholders consultation will elucidate which intervention elements are the most promising elements for future implementation in other clinics or countries [39].

## Conclusions

Participation in a multifaceted antimicrobial stewardship programme to optimise AMU in companion clinics, showed a positive effect on AMU in Dutch companion animal clinics. For future and feasible, large scale implementation, the most effective and efficient parts of the ASP need to be selected.

## Acknowledgments

We are very grateful to all 44 participating veterinary clinics for providing their AMU data and for investing their time during participation in the ASAP-project. We thank R.J. Wessels for his help during the post educational trainings and we thank A.W. van der Velden for her help in developing the patient evaluation forms. We also thank the remaining members of the ASAP-project group, L.J. Hellebrekers and M.F.M. Langelaar, for their highly-appreciated input.

## Author Contributions

**Conceptualization:** Marlies E. J. L. Hulscher, Dick J. J. Heederik, T. J. M. Verheij, Jaap A. Wagenaar, Jan M. Prins, Tjerk Bosje, Louska Schipper, Ingeborg M. van Geijlswijk, Els M. Broens.

**Formal analysis:** Nonke E. M. Hopman, Lützen Portengen, Dick J. J. Heederik.

**Project administration:** Nonke E. M. Hopman.

**Supervision:** Ingeborg M. van Geijlswijk, Els M. Broens.

**Writing – original draft:** Nonke E. M. Hopman, Dick J. J. Heederik, Jaap A. Wagenaar, Ingeborg M. van Geijlswijk, Els M. Broens.

**Writing – review & editing:** Nonke E. M. Hopman, Lützen Portengen, Marlies E. J. L. Hulscher, Dick J. J. Heederik, T. J. M. Verheij, Jaap A. Wagenaar, Jan M. Prins, Tjerk Bosje, Ingeborg M. van Geijlswijk, Els M. Broens.

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
