## [Decision Letter · Decision Letter 0]

8 Aug 2019

PONE-D-19-19385

Implementation and evaluation of an Antimicrobial Stewardship Programme in companion animal clinics: a stepped-wedge design intervention study

PLOS ONE

Dear Dr. Broens,

Thank you for submitting your manuscript to PLOS ONE. After careful consideration, we feel that it has merit but does not fully meet PLOS ONE’s publication criteria as it currently stands. Therefore, we invite you to submit a revised version of the manuscript that addresses the points raised during the review process.

We would appreciate receiving your revised manuscript by Sep 22 2019 11:59PM. To enhance the reproducibility of your results, we recommend that if applicable you deposit your laboratory protocols in protocols.io, where a protocol can be assigned its own identifier (DOI) such that it can be cited independently in the future. For instructions see: http://journals.plos.org/plosone/s/submission-guidelines#loc-laboratory-protocols

We look forward to receiving your revised manuscript.

Kind regards,

Simon Russell Clegg, PhD

Academic Editor

PLOS ONE

Additional Editor Comments (if provided):

Many thanks for submitting your manuscript to PLOS One

Your manuscript was reviewed by three expert reviewers who were generally very complimentary about your manuscript.

The majority of the comments are around grammatical and typographical issues. The reviewers spent a lot of time writing a very detailed response in order to aid your manuscript.

If you could write a response to reviewers, two of the same reviewers will then be invited to re-review the manuscript when resubmitted

I wish you good luck with your modifications

Many thanks

Simon

Reviewers' comments:

Reviewer's Responses to Questions

**Comments to the Author**

1. Is the manuscript technically sound, and do the data support the conclusions?

Reviewer #1: Yes

Reviewer #2: Yes

Reviewer #3: Yes

2. Has the statistical analysis been performed appropriately and rigorously? 

Reviewer #1: Yes

Reviewer #2: Yes

Reviewer #3: Yes

3. Have the authors made all data underlying the findings in their manuscript fully available?

Reviewer #1: Yes

Reviewer #2: No

Reviewer #3: No

4. Is the manuscript presented in an intelligible fashion and written in standard English?

Reviewer #1: Yes

Reviewer #2: Yes

Reviewer #3: Yes

5. Review Comments to the Author

Reviewer #1: This study assesses the impact of an antimicrobial stewardship program (ASP) on prescribing practices in 41 Dutch companion animal practices. ASP has been relatively neglected in companion animals globally compared to the focus on food animals, but is an important topic because people interact more closely with pet than with food animals (or their products). As expected from this group, the study was very well designed, in this case using a stepped-wedge intervention study implementing an ASP which was specifically developed for this study, and used practice management software to obtain the monthly data on antimicrobial use (AMU) needed (cats, dogs, rabbits) 2-years prior and during the 12-month ASP study period. The ASP program used a “RESET” model to guide changes in prescribing behaviour. The study showed a significant reduction in AMU and a significant shift to use of AMs of lower importance (except for third choice which were already low). There were considerable clinic differences. The discussion is excellent, self-critical of the limitations but with excellent arguments for the validity of the findings. The discussion is an excellent guide to others in other countries contemplating this type of needed action. This article leads the way in an on objective science-based intervention and analysis. The writing, tables, and figures are uniformly excellent. This is a very important paper in its field.

Line 292: net spelling

302: The goal of ….

Reviewer #2: The manuscript could be a good addition to scientific literature. It is generally well written except for a few issues that can be fixed. It might be beneficial to have an English language specialist read through (proof read). We do this quite often (have an English language specialist proof read our manuscripts) for our manuscripts. I am happy to review it again after the authors have addressed the issues that I raise.

Please see my comments below:

Abstract

Please consider indicating the time when the intervention study was conducted in the abstract.

Line 42-43: Please consider rephrasing...when you say to to perform and evaluate the effectiveness of an ASP ...perhaps you meant to say to implement and evaluate.... This also applies to lines 101-103.

Line 47: Should begin with...The objectives

Introduction

Line 71: Please mention some of the numerous elements of improved AMU.

Lines 74-76: Please consider rewriting this statement. Looks like a comma is missing before the word "between" in line 74.

Lines 76-77: Consider rephrasing.

Line 79: It would have been nice if reference 13 was in English. Can you try to translate reference 13 to English? Is there an English version? Having a link to an English version of reference 13 would benefit a reader who has no knowledge of Dutch. Just a thought. Please, provide a URL to reference 13.

Lines 86-87: This statement is either not well written or incomplete. For example "when comparing 2012-2014" is very confusing. Did you mean when comparing 2012 to 2014? Please consider rephrasing.

Line 100: Please consider adding "the" before "present".

Method

Line 104: Should it not be materials and methods? I believe you used some materials and several methods. Please consider replacing "Method" with "Materials and methods".

Line 118: Did you mean first cluster clinics and not "first clinics"? It is a bit confusing.

Line 120: Consider adding "The" before "clinics".

Line 129: Please consider replacing "from an earlier conducted study" with from a previous study. It might add value to also mention where this previous study was conducted.

Line 134: Please consider adding the word "The" before "clinics".

Line 141: Consider removing the word so-called (and throughout the manuscript). Depending on the reader, the word so-called may have a negative connotation. It may be interpreted that you think the use of such words are inappropriate. Please, check all the different dictionary meanings of the word so-called. You may choose to use "Mixed animal practices", if you believe "mixed clinics" is not appropriate.

Line 158: Please add "The" before "objectives".

Line 162: Delete "so-called".

Lines 164-167: I am wondering why there was no veterinary epidemiologist and/or infection prevention/control specialist in the S-team. I strongly believe that infection prevention/control is a very important element of an ASP. Any clarifications on this?

Line 168: Consider adding the word The before Dutch.

Line 171: I accessed references 25 and 26, but they are in Dutch. I wish there were English versions too.

Line 186: The primary outcome measure was total AMU. I look forward to the time when we will start to also consider metrics such as antibiotic associated length of stay in vet clinics,clinical response,AMR associated mortalities etc. These metrics have been suggested in human medicine (please read https://academic.oup.com/cid/article/59/suppl_3/S112/318184).

Line 187: Please consider moving reference 26 to line 168. consider inserting it after classification and before of.

Line 195: Presented for 24-12 months and 12-0 months (instead of 12-24 months and 0-12 months) may be confusing to an ordinary reader. Any reasons for this?

Lines 195-197: Please consider rephrasing the statement in lines 195-197 for clarity. In line 196, please insert the after to and before start.

Lines 198-199: This is really not a paragraph.

Line 201: Please, consider moving table 2 to just after table 1 and before data collection and management.

Results.

Lines 239-242: Did you mean to say that all clinics provided AMU data prior to the introduction of the ASP for a minimum of 24 months except for one clinic? Using the word "could" makes it sound like there is some doubt.. Please consider deleting the word could and rephrasing the statement in lines 239-242. Could in this context is used to express possibility (which could be uncertain possibility).

Line 244: Please, try to explain to the reader how the clinic got lost to follow-up. I believe there is a reasonable explanation for this loss to follow-up.

Lines 261-263: Please try rewriting this table heading for clarity. Make it simple for even an eighth grader.

Line 276: Did you mean Figures 2A-D? And not Figure 2. You have figures 2A-D, I didn't see a specific figure labeled Figure 2. Please clarify.

Discussion

Line 287: If you are sure of your findings, then you should unequivocally state that you attribute the observed shift to participation in the ASP.

Line 292: Replace nett with net. Also consider replacing effect is with effect was.

Line 296: Consider replacing "No statistical" with No statistically

Lines 297-301: Very long statement. Consider rephrasing.

Lines 300-301: when you say...due to lack of statistical power, are you not negating your power calculation in lines 125-129? Please clarify or rephrase.

Line 302: Insert "The" before "Goal".

Lines 304-305: Please check the statement beginning with "The observed..." for grammar/syntax.

Lines 313-325: This is a good discussion. This paper "" ext-link-type="uri" xlink:type="simple">https://academic.oup.com/cid/article/59/suppl_3/S112/318184" may also be useful for your discussion.

Line 344: Please consider replacing irrefutable with irrefutably.

Line 348: Replace AM with Antimicrobial.

Line 353: Please add "a" after as and before factor.

Lines 360-365: I wonder how useful this paragraph is to the paper. Lines 362-364 is confusing. Please consider rewriting in a more clear language.

Lines 366-370: In my view, this discussion points are not necessary for this manuscript.

Conclusion

The presented conclusion appears too general.

Lines 375-376: From your study, please mention the most effective and efficient parts of the ASP that need to be selected.

Reviewer #3: This is an excellent paper and well worthy of publication. The statistics appear to be rigorously performed and the sample size and intervention achieved mean the results have real impact for the veterinary community.

Line 79-80. This sentence needs rewording to improve clarity. Does the Dutch vet association promote the use of guidelines or require them???

Line 88-92: This needs a reference.

6. PLOS authors have the option to publish the peer review history of their article (what does this mean?). If published, this will include your full peer review and any attached files.

Reviewer #1: No

Reviewer #2: No

Reviewer #3: No

---

## [Author Response · Author response to Decision Letter 0]

8 Oct 2019

This information is also included in the uploaded document named "Response to reviewers":

Review Comments to the Author

Reviewer #1: This study assesses the impact of an antimicrobial stewardship program (ASP) on prescribing practices in 41 Dutch companion animal practices. ASP has been relatively neglected in companion animals globally compared to the focus on food animals, but is an important topic because people interact more closely with pet than with food animals (or their products). As expected from this group, the study was very well designed, in this case using a stepped-wedge intervention study implementing an ASP which was specifically developed for this study, and used practice management software to obtain the monthly data on antimicrobial use (AMU) needed (cats, dogs, rabbits) 2-years prior and during the 12-month ASP study period. The ASP program used a “RESET” model to guide changes in prescribing behaviour. The study showed a significant reduction in AMU and a significant shift to use of AMs of lower importance (except for third choice which were already low). There were considerable clinic differences. The discussion is excellent, self-critical of the limitations but with excellent arguments for the validity of the findings. The discussion is an excellent guide to others in other countries contemplating this type of needed action. This article leads the way in an on objective science-based intervention and analysis. The writing, tables, and figures are uniformly excellent. This is a very important paper in its field.

Line 292: net spelling

302: The goal of ….

Dear Reviewer 1, thank you very much for critically revising our manuscript and your very complimentary words. We changed the word “nett” into “net” (Line 303) and “Goal of the ASP” was changed into “The Goal of the ASP” (Line 312). 

Reviewer #2: The manuscript could be a good addition to scientific literature. It is generally well written except for a few issues that can be fixed. It might be beneficial to have an English language specialist read through (proof read). We do this quite often (have an English language specialist proof read our manuscripts) for our manuscripts. I am happy to review it again after the authors have addressed the issues that I raise.

Dear Reviewer 2, thank you very much for critically revising our manuscript, your kind words and the given opportunity to improve our manuscript. We will address all the issues you raise separately below. 

Please see my comments below:

Abstract

Please consider indicating the time when the intervention study was conducted in the abstract. 

The text “which was performed from March 2016 until March 2018” is added now in Line 37. 

Line 42-43: Please consider rephrasing...when you say to to perform and evaluate the effectiveness of an ASP ...perhaps you meant to say to implement and evaluate.... This also applies to lines 101-103.

“Perform” was replaced by “implement” in both sentences, Line 34 and 95.

Line 47: Should begin with...The objectives

“The” is added in Line 39.

Introduction

Line 71: Please mention some of the numerous elements of improved AMU.

“(e.g., increasing awareness of (inter)national practice guidelines, use of diagnostic microbiology and use of alternatives to AMs)” was added to mention some examples of the numerous elements of improved AMU, Line 65-66.

Lines 74-76: Please consider rewriting this statement. Looks like a comma is missing before the word "between" in line 74.

This sentence was rephrased: “In Dutch food producing animals, a combination of compulsory and voluntary actions resulted in a 64% reduction in AMU (between 2008 and 2017). A decrease in resistance rates was observed as well.” Line 68-70.

Lines 76-77: Consider rephrasing.

This sentence was rephrased: “Just since the end of 2011 onwards, more attention is being paid to AMU in companion animals”, Line 70-71.

Line 79: It would have been nice if reference 13 was in English. Can you try to translate reference 13 to English? Is there an English version? Having a link to an English version of reference 13 would benefit a reader who has no knowledge of Dutch. Just a thought. Please, provide a URL to reference 13. 

As it concerns Dutch law, the reference is in Dutch and there is no English version of it. We added an URL to the Dutch website and we translated the title into English (“Regulation of the State Secretary of Economic Affairs”) in the reference list, but unfortunately there is no English version. 

Lines 86-87: This statement is either not well written or incomplete. For example "when comparing 2012-2014" is very confusing. Did you mean when comparing 2012 to 2014? Please consider rephrasing.

This sentence was rephrased: “During the past years, AMU in Dutch companion animal clinics has been decreasing (with 19% when comparing 2012 to 2014)”, Line 80-81.

Line 100: Please consider adding "the" before "present".

“The” was added in Line 94.

Method

Line 104: Should it not be materials and methods? I believe you used some materials and several methods. Please consider replacing "Method" with "Materials and methods".

Thank you, you are completely right. The heading is now “Materials and methods”. 

Line 118: Did you mean first cluster clinics and not "first clinics"? It is a bit confusing.

First clinics was changed into “The first cluster of clinics was contacted”, Line 112.

Line 120: Consider adding "The" before "clinics".

“The” was added in Line 114. 

Line 129: Please consider replacing "from an earlier conducted study" with from a previous study. It might add value to also mention where this previous study was conducted.

The sentence was changed as you suggested: “Calculations were based on AMU data at clinic level from a previous study conducted in 68 Dutch companion animal clinics”, Line 127-128. 

Line 134: Please consider adding the word "The" before "clinics".

“The” was added before clinics, Line 132. 

Line 141: Consider removing the word so-called (and throughout the manuscript). Depending on the reader, the word so-called may have a negative connotation. It may be interpreted that you think the use of such words are inappropriate. Please, check all the different dictionary meanings of the word so-called. You may choose to use "Mixed animal practices", if you believe "mixed clinics" is not appropriate.

Thank you very much for this comment and your explanation. “So-called” was removed in Line 139 and in Line 160. 

Line 158: Please add "The" before "objectives".

“The” was added, Line 156.

Line 162: Delete "so-called".

“So-called” was deleted, in Line 160. 

Lines 164-167: I am wondering why there was no veterinary epidemiologist and/or infection prevention/control specialist in the S-team. I strongly believe that infection prevention/control is a very important element of an ASP. Any clarifications on this?

Thank you very much for this comment. We fully agree that infection prevention/control is very important in all strategies to preserve AMs and in A-teams in human hospitals an infection prevention/control specialist is involved. In the ASAP-project group an epidemiologist (also in author list) and an infection prevention/control specialist were involved. But indeed, not directly in the S-team. One of the explanations hereof has to do with logistic issues. Another explanation is that our ASP was primarily focused on prudent use of AMs (right choice, dose and length) and less on infection prevention. Line 168: Consider adding the word The before Dutch. 

“The” was added, Line 166.

Line 171: I accessed references 25 and 26, but they are in Dutch. I wish there were English versions too.

Unfortunately, only Dutch versions of these documents exist, no English translations are available. As the Dutch classification of AMs (reference 26) is crucial for the interpretation of our data/results, this is explained in Table 2. 

Line 186: The primary outcome measure was total AMU. I look forward to the time when we will start to also consider metrics such as antibiotic associated length of stay in vet clinics,clinical response,AMR associated mortalities etc. These metrics have been suggested in human medicine (please read https://academic.oup.com/cid/article/59/suppl_3/S112/318184).

Thank you for this comment. We totally agree and we therefore mention this in the discussion. 

Line 187: Please consider moving reference 26 to line 168. consider inserting it after classification and before of.

Reference 26 is moved to Line 166 and inserted before “of”. 

Line 195: Presented for 24-12 months and 12-0 months (instead of 12-24 months and 0-12 months) may be confusing to an ordinary reader. Any reasons for this?

We understand the potential confusion of the used notation. We changed it into 12-24 and 0-12, both in Line 200 and Table 4. 

Lines 195-197: Please consider rephrasing the statement in lines 195-197 for clarity. In line 196, please insert the after to and before start.

As you suggested, the statement was rephrased and in Line 201 “the” was added. 

Lines 198-199: This is really not a paragraph.

We now placed this sentence in the previous paragraph, Line 197-198. 

Line 201: Please, consider moving table 2 to just after table 1 and before data collection and management.

Table 2 was moved as you suggested. 

Results.

Lines 239-242: Did you mean to say that all clinics provided AMU data prior to the introduction of the ASP for a minimum of 24 months except for one clinic? Using the word "could" makes it sound like there is some doubt.. Please consider deleting the word could and rephrasing the statement in lines 239-242. Could in this context is used to express possibility (which could be uncertain possibility).

Thank you, it was indeed as you suggested, all clinics provided “before” data of at least 24 months, except for one clinic. Therefore the sentence was changed: “All clinics provided AMU data prior to the introduction of the stewardship programme for a minimum of 25 months, except for one clinic, that provided data for only 13 months prior to the introduction of the ASP”, Line 239-242. 

Line 244: Please, try to explain to the reader how the clinic got lost to follow-up. I believe there is a reasonable explanation for this loss to follow-up. 

This clinic was lost to follow-up, because the clinic did not respond anymore to repeated phone calls and e-mails, when trying to retrieve their AM prescription data after participation in the ASP (which they did fully participate in). Therefore, this line was added: “One of these clinics was lost to follow-up (i.e. no response was received when (repeatedly) trying to retrieve the AM prescription data from the PMS after participation in the ASP) ”, Line 244-246. 

Lines 261-263: Please try rewriting this table heading for clarity. Make it simple for even an eighth grader.

The table heading was rewritten: “Mean total, 1st, 2nd and 3rd choice AMU (in numbers of DDDA/month and percentage of total AMU) in participating clinics, before and during participation in the antimicrobial stewardship programme (ASP). 

Line 276: Did you mean Figures 2A-D? And not Figure 2. You have figures 2A-D, I didn't see a specific figure labeled Figure 2. Please clarify.

You are right, I indeed mean Fig. 2A-D. This was changed. 

Discussion

Line 287: If you are sure of your findings, then you should unequivocally state that you attribute the observed shift to participation in the ASP.

As we are sure of our findings, we rephrased this sentence: “that was attributed to participation in an antimicrobial stewardship programme” (Line 298-299). 

Line 292: Replace nett with net. Also consider replacing effect is with effect was.

Both changes were made: “the net effect was a less pronounced increase”, Line 303. 

Line 296: Consider replacing "No statistical" with No statistically

“No statistical” was replaced with “No statistically”, Line 307. 

Lines 297-301: Very long statement. Consider rephrasing. 

Lines 300-301: when you say...due to lack of statistical power, are you not negating your power calculation in lines 125-129? Please clarify or rephrase.

This sentence was rephrased: “This is likely explained by the fact that 3rd choice AMU was already reduced to a low level (0.009 DDDA/month, 0-12 months before implementation of the ASP) in the years preceding the ASP. Therefore, a further decrease as result of the ASP is difficult to demonstrate. 

Line 302: Insert "The" before "Goal".

“The” was inserted, in Line 312. 

Lines 304-305: Please check the statement beginning with "The observed..." for grammar/syntax.

This sentence was changed: “The observed changes in the clinics participating in the ASP are in line with the goal of the ASP and the Dutch guidelines on veterinary AMU, and are therefore considered relevant”, Line 314-316.

Lines 313-325: This is a good discussion. This paper "https://academic.oup.com/cid/article/59/suppl_3/S112/318184" may also be useful for your discussion.

Thank you for this compliment and the reference. The suggested paper is used in the discussion now, Line 324-326. “In general, measures to evaluate the effect of an ASP can be divided into four main categories: patient outcomes, unintended consequences of AMU (e.g., adverse effects and emergence of AMR), AMU and costs, and process measures [29]”. And it was added as reference in Line 329.

Line 344: Please consider replacing irrefutable with irrefutably.

“Irrefutable” is replaced with “irrefutably”, Line 358. 

Line 348: Replace AM with Antimicrobial.

AM was replaced with “antimicrobial”, Line 362.

Line 353: Please add "a" after as and before factor.

In Line 367 “a” was added. 

Lines 360-365: I wonder how useful this paragraph is to the paper. Lines 362-364 is confusing. Please consider rewriting in a more clear language.

We rewrote his paragraph. Until now we only evaluated the effectiveness of the ASP on AMU, we did not evaluate the opinions and experiences of the participants yet. This is necessary to answer which intervention element(s) worked best. “The present study showed that the developed ASP was effective in reducing and refining AMU in the participating clinics. An evaluation survey among participants and a stakeholders consultation will elucidate which intervention elements are the most promising elements for future implementation in other clinics or countries [39]” (Line 374-377). 

Lines 366-370: In my view, this discussion points are not necessary for this manuscript.

We agree with your comment and therefore omitted this discussion point. 

Conclusion

The presented conclusion appears too general.

Lines 375-376: From your study, please mention the most effective and efficient parts of the ASP that need to be selected.

Until now, it was not possible yet to mention the most effective and efficient parts of the ASP, this should be based upon an evaluation with the participants. Therefore, this will soon be our next step to enable future and feasible, large scale implementation. 

Reviewer #3: This is an excellent paper and well worthy of publication. The statistics appear to be rigorously performed and the sample size and intervention achieved mean the results have real impact for the veterinary community.

Dear Reviewer, thank you very much for reviewing our manuscript and for your kind words. 

Line 79-80. This sentence needs rewording to improve clarity. Does the Dutch vet association promote the use of guidelines or require them???

This sentences was clarified as you suggested: “The Dutch Veterinary Association “promotes” the use of guidelines on AMU as well”, Line 73-74. 

Line 88-92: This needs a reference. 

A reference is now added in Line 86.

---

## [Decision Letter · Decision Letter 1]

30 Oct 2019

Implementation and evaluation of an Antimicrobial Stewardship Programme in companion animal clinics: a stepped-wedge design intervention study

PONE-D-19-19385R1

Dear Dr. Broens

We are pleased to inform you that your manuscript has been judged scientifically suitable for publication and will be formally accepted for publication once it complies with all outstanding technical requirements.

With kind regards,

Simon Russell Clegg, PhD

Academic Editor

PLOS ONE

Additional Editor Comments (optional):

Many thanks for resubmitting your interesting manuscript to PLOS One, and for the time taken to produce a good response to reviewers

The manuscript was re-reviewed by the same reviewers as before, and I am pleased to say that they have recommended that the manuscript be accepted for publication

I wish you all the best for your future research

Many thanks

Simon

Reviewers' comments:

Reviewer's Responses to Questions

**Comments to the Author**

1. If the authors have adequately addressed your comments raised in a previous round of review and you feel that this manuscript is now acceptable for publication, you may indicate that here to bypass the “Comments to the Author” section, enter your conflict of interest statement in the “Confidential to Editor” section, and submit your "Accept" recommendation.

Reviewer #2: All comments have been addressed

Reviewer #3: All comments have been addressed

2. Is the manuscript technically sound, and do the data support the conclusions?

Reviewer #2: Yes

Reviewer #3: Yes

3. Has the statistical analysis been performed appropriately and rigorously? 

Reviewer #2: Yes

Reviewer #3: Yes

4. Have the authors made all data underlying the findings in their manuscript fully available?

Reviewer #2: No

Reviewer #3: No

5. Is the manuscript presented in an intelligible fashion and written in standard English?

Reviewer #2: Yes

Reviewer #3: Yes

6. Review Comments to the Author

Reviewer #2: (No Response)

Reviewer #3: (No Response)

7. PLOS authors have the option to publish the peer review history of their article (what does this mean?). If published, this will include your full peer review and any attached files.

Reviewer #2: No

Reviewer #3: No

---

## [Editor Report · Acceptance letter]

8 Nov 2019

PONE-D-19-19385R1 

Implementation and evaluation of an Antimicrobial Stewardship Programme in companion animal clinics: a stepped-wedge design intervention study 

Dear Dr. Broens:

I am pleased to inform you that your manuscript has been deemed suitable for publication in PLOS ONE. Congratulations! Your manuscript is now with our production department. 

With kind regards,

on behalf of

Dr. Simon Russell Clegg 

Academic Editor

PLOS ONE